ᵃ | **Open Peer Review** | Bacteriology | Methods and Protocols

# ESKAPE Gram-negative bacteria escape culture-based detection upon desiccation on abiotic surfaces

Daniela Visaggio,[1,2,3] Massimiliano Lucidi,[1,2] Cinzia Spagnoli,[1] Ilaria Ciccone,[1] Francesco Imperi,[1,2,3] Paolo Visca[1,2,3]

**ABSTRACT**   ESKAPE bacteria, namely *Enterococcus* spp., *Staphylococcus aureus*, *Klebsiella pneumoniae*, *Acinetobacter baumannii*, *Pseudomonas aeruginosa*, and *Enterobacter* spp., are leading causes of hospital-acquired infections and a major therapeutic challenge due to multidrug resistance. Hospital surfaces and medical devices are critical reservoirs for the transmission of these pathogens to patients. Standard methods for detecting microorganisms in the hospital environment are culture-based, so they cannot identify bacteria in the viable but non-culturable (VBNC) state. VBNC bacteria remain metabolically active and potentially infectious, but they fail to grow in conventional, nutrient-rich culture media. Reversion from the VBNC to the cultivable state is termed resuscitation. To assess whether ESKAPE species enter the VBNC state upon desiccation on abiotic materials commonly utilized in clinical facilities and can be resuscitated, bacterial cells were desiccated for 1 week on glass, different plastics, cotton, and titanium surfaces, then resuscitated in a carbon-free buffer. After desiccation, all ESKAPE pathogens exhibited reduced cultivability, with species- and surface-dependent variability. Gram-positive ESKAPE species did not regain cultivability after resuscitation. Conversely, Gram-negative species reverted to the cultivable state, indicating a transition to the VBNC state in response to desiccation. Compared to the standard methodology for biocontamination control (EN 17141:2020), the resuscitation step prior to culture yielded a significantly greater recovery of Gram-negative ESKAPE bacteria in the VBNC state from both experimentally contaminated samples and environmental surfaces. These findings pose the need for environmental monitoring approaches capable of detecting VBNC pathogens on abiotic hospital surfaces.

**IMPORTANCE** Accurate detection of microbial contamination in the hospital environment is fundamental for preventing nosocomial infections. Current protocols for environmental surveillance, however, rely almost exclusively on culture-based methods, which overlook bacteria in the viable but non-culturable (VBNC) state. This study demonstrates that clinically relevant Gram-negative ESKAPE pathogens can persist on hospital surfaces in the VBNC state, thereby evading conventional approaches for environmental control, resulting in substantial underestimation of the bacterial burden. We further show that a simple resuscitation step restores the cultivability of VBNC cells, improving their recovery rate, ultimately resulting in much greater sensitivity compared with conventional biocontamination control methods. These findings reveal a critical limitation of current environmental surveillance approaches and highlight the importance of integrating VBNC detection into monitoring protocols for achieving a more accurate assessment of surface contamination to strengthen infection prevention strategies.

**KEYWORDS**   biocontamination, ESKAPE, hospital environment, resuscitation, sampling, VBNC

**Peer Reviewer** Hajime Kanamori, University of North Carolina at Chapel Hill, Chapel Hill, North Carolina, USA

Address correspondence to Daniela Visaggio, daniela.visaggio@uniroma3.it, or Paolo Visca, paolo.visca@uniroma3.it.

Daniela Visaggio and Massimiliano Lucidi contributed equally to this article. The author order was determined by decreasing seniority and contribution to funding acquisition.

The authors declare no known conflict of interest or personal relationships that could have appeared to influence the work reported in this paper. A patent entitled "Metodo per la quantificazione di microrganismi patogeni su superfici abiotiche" was filed on 19 June 2025, application number: 102025000014596, Italian Patent Office (Ufficio Italiano Brevetti e Marchi).

See the funding table on p. 10.

Hospital surfaces and medical devices are recognized as critical sources for pathogen transmission, facilitating cross-contamination of patients through various vectors, including healthcare personnel and medical equipment (1, 2). Although monitoring microbial contamination of hospital surfaces is crucial for containing an outbreak of infection when an environmental reservoir is suspected, contamination of the hospital environment is rarely investigated, and internationally agreed methodological guidelines are unavailable (3–5). Environmental sampling procedures in healthcare facilities may involve several methods and different devices (swabs, sponges, contact plates), but invariably rely on culture-based approaches, overlooking the ability of some bacterial species to enter a viable but non-culturable (VBNC) state (6–8). The VBNC state is a reversible condition in which viable bacterial cells lose cultivability upon exposure to an environmental stress. Reversion from VBNC to the culturable state is called resuscitation and occurs under conditions that vary between bacterial species and the type of stress (6, 9, 10).

In the hospital environment, bacterial contaminants of inanimate surfaces are exposed to multiple stresses, among which prolonged desiccation is particularly challenging. The progressive loss of water over extended periods of dryness threatens their survival, inducing biochemical, metabolic, and physiological changes (11).

Evidence has previously been provided that *Acinetobacter baumannii* strains lose cultivability after desiccation on abiotic surfaces by entering the VBNC state and can resuscitate when rehydrated in suitable buffers or biological fluids (12, 13). Therefore, transition to the VBNC state allows *A. baumannii* to persist on hospital surfaces as a concealed source of contamination, turning the inanimate environment into a hidden reservoir of infection. *A. baumannii* and other ESKAPE pathogens (*Enterococcus* spp., *Staphylococcus aureus, Klebsiella pneumoniae, Pseudomonas aeruginosa,* and *Enterobacter* spp.) are among the most dreaded bacteria responsible for healthcare-associated infections (HAI), being notorious for their high level of resistance to several antibiotics (14, 15). While ESKAPE pathogens have developed adaptations for survival in the modern health care setting (16) and are known to enter a VBNC state in response to several types of stress (Table S1), little is known about their behavior under desiccation stress. Here, we demonstrate that all Gram-negative ESKAPE bacteria can withstand desiccation by entering the VBNC state, and we propose a new environmental monitoring method that detects VBNC cells as well.

## MATERIALS AND METHODS

### Bacterial strains, sample preparation, and desiccation conditions

The bacterial strains used in this study are described in Table S2. Bacteria were pre-cultured in Luria Bertani broth (LB) at 37°C for 18 h, diluted 1:100 in fresh LB, and incubated at 37°C under shaking for 6 h. Bacterial cultures were harvested by centrifugation (3,000 × $g$ for 5 min), washed twice with double-distilled water, and suspended in double-distilled water at an optical density at 600 nm ($OD_{600}$) of 1.0. Twenty-microliter aliquots of bacterial suspensions in double-distilled water were deposited onto 1.5 cm$^2$ surfaces of glass (GL), polyvinyl chloride (PVC), polypropylene (PP), polyester (PL), polystyrene (PS), polyethylene (PE), silicon (SL), cotton (CT), and titanium (Ti) and air-dried under the laminar flow hood at 25.2°C ± 1.5°C. These materials are broadly used in hospital settings as outlined in Table S3. After air-drying, bacterial cells were stored for 1 week in a 16 L-vacuum bell containing 50 g of silica gel at an average temperature and relative humidity of 20.9°C ± 0.6°C and 13.0% ± 5.6%.

### Bacterial resuscitation conditions

To assess bacterial cultivability after desiccation, air-dried samples were rehydrated in 2 mL of resuscitation buffer (RB, 6.8 g/L $Na_2HPO_4$, 3 g/L $KH_2PO_4$, 0.5 g/L NaCl, 1 g/L $NH_4Cl$, 0.02 g/L $CaCl_2$, and 0.12 g/L $MgSO_4$; Table S4), incubated for 15 min at room

temperature, mixed by vortexing for 30 s, and 10-fold serially diluted in sterile saline for colony-forming unit (CFU) counts, as previously described (13). For resuscitation, the same tube from which the aliquot had been removed for the CFU counts was incubated at 37°C for 24 h under shaking, and CFU counts were performed on tryptic soy agar (TSA) plates.

## Effect of different swabs and transport buffers on bacterial cultivability after desiccation

Fifty-microliter aliquots of bacterial suspensions prepared as described above were poured onto glass slides and desiccated for 1 week. Bacterial cells were harvested using different swabs (Table S4). The nylon swab (Copan) was placed in 5 mL of RB, whereas the commercial swabs (Table S4) were placed in 5 mL of their respective transport buffers (TB, Table S4), except for the LMS Amies swab, which contained 1 mL of Amies solution. All swabs were incubated for 15 min at room temperature, mixed by vortexing for 30 s, and the bacterial suspensions were 10-fold serially diluted in sterile saline for CFU counts, in order to determine the bacterial cultivability immediately after desiccation. Swabs placed in RB were incubated at 37°C for 24 h under shaking, whereas the commercial ones were maintained at 4°C for 24 h under static conditions. CFU counts were then performed on TSA plates to assess the bacterial cultivability. The resuscitation in Amies buffer was also evaluated by incubating the samples at 37°C under shaking for 24 h.

## LIVE/DEAD staining for quantification of bacterial membrane integrity

Fluorescent labeling was performed using the LIVE/DEAD BacLight Bacterial Viability Kit (ThermoFisher Scientific), on bacterial suspensions before desiccation, after desiccation on glass Petri dishes, and after resuscitation, as previously described (13). The LIVE/DEAD BacLight Bacterial Viability Kit contains two nucleic acid dyes: SYTO 9, which penetrates all bacterial cells and fluoresces green upon DNA binding, and propidium iodide (PI), which enters only cells with damaged membranes, quenches the SYTO 9 emission, and emits red fluorescence. As PI quenches SYTO 9 fluorescence in non-viable cells, the green/red fluorescence ratio serves as a proxy for membrane integrity.

For microscopy, 20 µL of each stained suspension was deposited onto a glass coverslip coated with 0.5% (wt/vol) agarose to immobilize bacterial cells. Samples were imaged using a Nikon A1R HD25 confocal laser scanning microscope equipped with an Apo TIRF 100× oil immersion objective (numerical aperture = 1.49). For each condition, five images were acquired at a resolution of $512 \times 512$ pixels. Image analysis was conducted in ImageJ (Fiji) by manually selecting individual cells using the Multipoint Tool, adding the selected points to the ROI Manager, and exporting the pixel intensity values for red (R) and green (G) channels using a custom macro script (reported in Supplemental material).

Five randomly selected areas without cells were used to determine the background RG intensity ($RG_{background}$). For each cell, the green/red pixel intensity ratio was calculated and corrected by subtracting $RG_{background}$. Clusters in which single cells could not be clearly distinguished were excluded from the analysis. For each sample, >300 cells were analyzed.

## Growth of ESKAPE bacteria in RB supplemented or not with a nutrient-rich medium

Bacterial strains were grown in LB at 37°C for 18 h. An aliquot of the bacterial cultures was washed and diluted to $OD_{600} = 0.01$ in a final volume of 200 µL of RB. Alternatively, an aliquot was diluted to $OD_{600} = 0.02$ in 100 µL of 2×RB. Bacterial cultures were incubated at 37°C, and growth was monitored by recording the $OD_{600}$ every 6 h for 24 h using a Spark 10 M microplate reader (Tecan). After 24 h, wells containing 100 µL of bacterial cultures in 2×RB were supplemented with 100 µL of 2×LB. Bacterial cultures were incubated at 37°C, and growth was monitored for additional 24 h.

## Growth of ESKAPE bacteria in different transport buffers

Bacterial strains were grown in LB at 37°C for 6 h. An aliquot of the bacterial cultures was washed in double-distilled water and diluted to $OD_{600}$ = 0.001 in a final volume of 100 µL of D/E neutralizing broth (NB), maximum recovery diluent (MRD), neutralizing rinse solutions (NRS), and Amies solution. Bacterial cultures were incubated at 37°C, and growth was monitored by recording the $OD_{600}$ every 4 h for 24 h using a Spark 10 M microplate reader (Tecan).

## Environmental surface sampling

Laboratory benches, washbasins, mobile phones, and classroom desks were sampled using nylon swabs (Copan) pre-moistened in RB, LMS Swab Amies, or ESC Swab Maximum Recovery (Table S4). A 10 × 10 cm sampling mask (Copan) was used for all surfaces except for mobile phones, which were sampled using a 4 × 5 cm mask (Copan). The nylon swabs were placed in 2 mL of RB, whereas each commercial swab was placed in its respective buffer (Table S4). Samples were incubated at room temperature for 15 min, vortexed for 1 min, and 100-µL aliquots were plated on TSA plates for CFU counts. Then, swabs immersed in RB were incubated at 37°C with shaking for 24 h, while those in Amies and MRD were stored at 4°C for 24 h. After incubation, serial dilutions in sterile saline solution were prepared, and 100 µL aliquots were plated on TSA plates for CFU counts. The plates were incubated for 72 h at 37°C. For each surface type, six samples were collected and analyzed using nylon swab and RB, while three samples each were collected and analyzed using the LMS swab Amies and ESC Swab Maximum Recovery.

## Statistical analysis

Statistical analysis was performed using the GraphPad Instat software v8.0. Data were analyzed by Student's *t*-test and one-way ANOVA, as indicated in the figure legends. Asterisks indicate the statistical significance: *$P < 0.05$, **$P < 0.01$, ***$P < 0.001$, ****$P < 0.0001$, ns = not significant.

## RESULTS

## Gram-negative ESKAPE bacteria survive desiccation by entering the VBNC state

Entry into a VBNC state in response to desiccation stress has only been reported for *A. baumannii* (13). This trait may enable bacteria to persist for extended periods and eventually spread in the hospital environment, thereby increasing the likelihood of patient exposure and the risk of healthcare-associated infections. Initially, we wondered whether the ability to withstand desiccation stress by entering the VBNC state is a common feature of ESKAPE pathogens. To this aim, one clinical isolate and one reference strain of ESKAPE species (Table S2) were desiccated for 1 week at room temperature on the surfaces of different materials frequently found in hospitals, namely glass, plastic polymers, cotton, and titanium (Table S3; Fig. 1A). The CFU counts revealed that cultivability after 1 week of desiccation varied among strains and was influenced by the type of material (Fig. 1B; Fig. S1). All ESKAPE bacteria, except *K. pneumoniae* and *A. baumannii*, exhibited a significant reduction in cultivability upon dehydration on almost all surfaces. This reduction was particularly pronounced for *P. aeruginosa* strains, as well as for clinical *Klebsiella aerogenes* (basonym *Enterobacter aerogenes*), and *S. aureus* isolates (Fig. 1B; Fig. S1 and S2). To differentiate cell death from entrance into the VBNC state, a resuscitation experiment was conducted. One-week desiccated bacterial cells were incubated for 24 h at 37°C in an isotonic buffer, named RB, which does not support bacterial growth due to the lack of any carbon sources (13) (Table S4). As expected, the absence of carbon sources impeded bacterial growth in RB (Fig. S3).

All Gram-negative bacteria fully or partly recovered cultivability after resuscitation, depending on the tested material (Fig. 1B; Fig. S1), denoting their transition to a desiccation-induced VBNC state, which was completely or largely reversed to a culturable

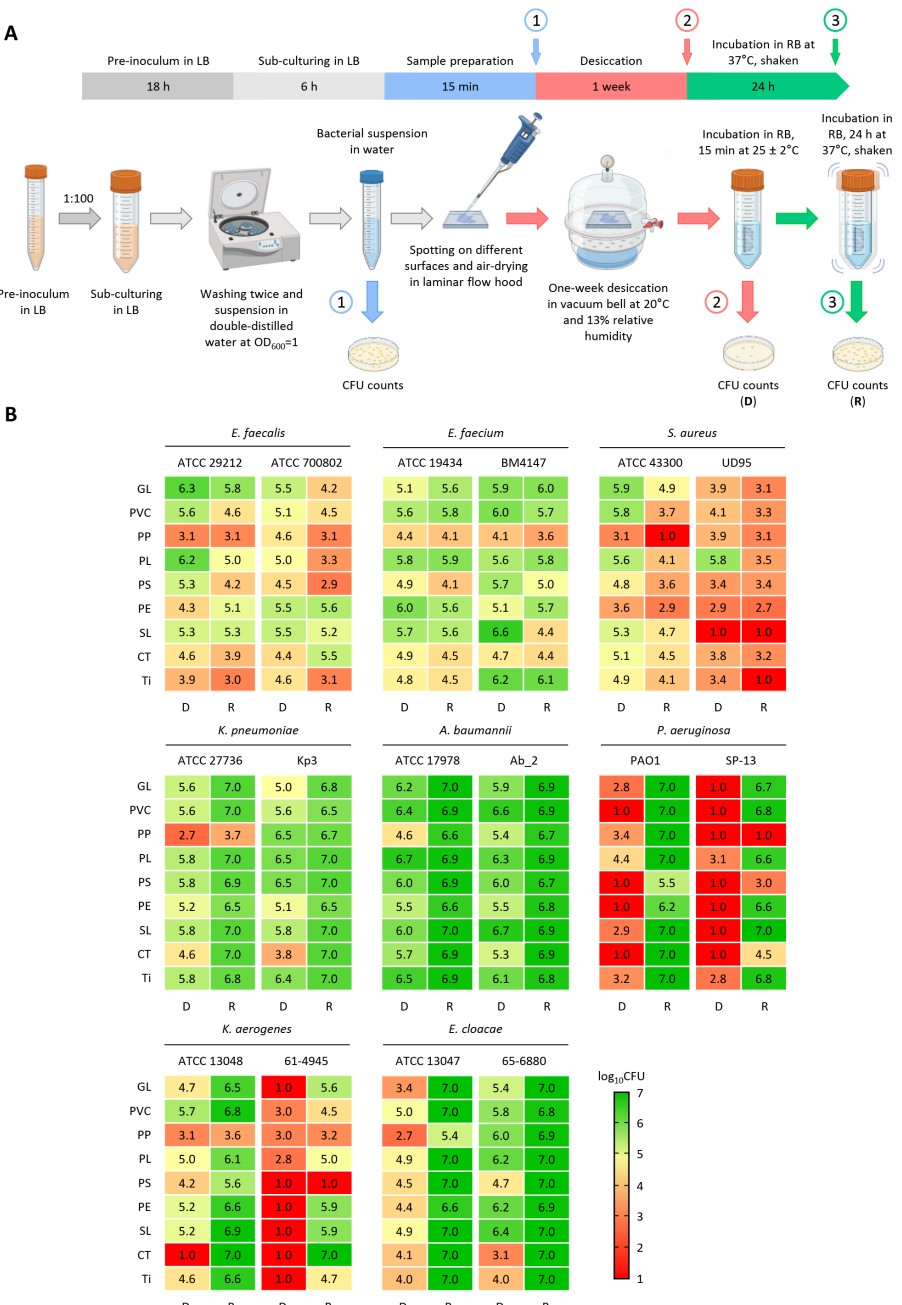

**FIG 1** Cultivability of ESKAPE bacteria after desiccation on different surfaces and subsequent resuscitation. (A) Experimental timeline and schematization of desiccation resistance assay. The experiment comprises four sequential steps: (i) bacterial cultivation (gray), (ii) sample preparation (blue), (iii) desiccation (red), and (iv) resuscitation (green). The numbered arrows indicate the time points at which CFU counts were performed, i.e., before desiccation (1), after 1-week desiccation on different surfaces (2), and after resuscitation in RB at 37°C for 24 h under shaking (3). Panel A was created with Biorender. (B) Cultivability, expressed as $\log_{10}$ CFU, after 1-week desiccation (D lane) and after resuscitation in RB (R lane). Data are the means of three independent experiments. Mean values and error bars are shown in Fig. S1. Abbreviations of surfaces: glass (GL), polyvinyl chloride (PVC), polypropylene (PP), polyester (PL), polystyrene (PS), polyethylene (PE), silicon (SL), cotton (CT), and titanium (Ti).

state upon rehydration in RB. Conversely, the recovery of CFUs after incubation in RB was either equal to or lower than that observed after 1-week desiccation for all Gram-positive ESKAPE species (Fig. 1B; Fig. S1), suggesting that desiccation does not induce the

VBNC state in these pathogens, or the experimental conditions were unsuitable for their resuscitation.

To assess membrane damage following dehydration, cells were stained with the LIVE/DEAD dyes before desiccation, after 1-week desiccation, and after resuscitation. With the exception of *Enterococcus faecium* and *Enterococcus faecalis*, whose desiccated cells were not permeable to SYTO 9 (Fig. S4), all ESKAPE species showed significant loss of membrane integrity after desiccation (Fig. S4). After resuscitation, partial or no restoration of membrane integrity was observed in most species, with the only exception of *P. aeruginosa*, which displayed a nearly complete recovery of membrane integrity (Fig. S4). Therefore, no consistent correlation between the membrane integrity and cultivability was observed for almost all ESKAPE bacteria, indicating that Gram-negative ESKAPE bacteria can regain cultivability even without evident repair of the desiccation-induced membrane damage.

## Bacteria in the VBNC state escape standard detection methods

According to the BS EN 17141:2020 standard procedure suited for "Cleanrooms and associated controlled environments—Biocontamination control", after surface sampling with a swab moistened in a suitable TB, samples must be transported at a temperature between 1°C and 8°C. Then, an aliquot of the TB should be directly plated on TSA or stored at 3°C ± 2°C for no longer than 48 h before plating. Since Gram-negative ESKAPE bacteria enter the VBNC state upon desiccation, the BS EN 17141:2020 standard procedure was assessed in comparison with the resuscitation protocol for VBNC bacteria described in the previous section. Bacterial cells were desiccated on a glass slide for 1 week and harvested using a nylon swab soaked in RB or four types of commercial swabs soaked in the cognate TB (Table S4). The swab soaked in RB was incubated in 5 mL RB at 37°C for 24 h under shaking, whereas, according to the BS EN 17141:2020 recommendations, the commercial swabs were kept at 4°C under static conditions for 24 h in 5 mL of TB (Fig. 2A), except the LMS Amies swab, which contains 1 mL of Amies medium. Substantial differences were observed when CFUs were quantified after 24-h incubation in the different buffers (Fig. 2B; Fig. S5).

Regarding Gram-positive ESKAPE bacteria, incubation in RB resulted in a slight reduction of CFU counts compared to the commercial swabs, except for *E. faecium* ATCC 19436, which exhibited a moderate increase in CFU counts. Conversely, all Gram-negative ESKAPE bacteria were completely resuscitated in RB, but not in any of the TBs (Fig. S5). These results indicate that standard sampling methods fail to detect VBNC cells, leading to an underestimation of the number of viable bacteria on surfaces.

To verify whether bacterial resuscitation in a commercial buffer is influenced by incubation temperature, 1-week desiccated bacteria were harvested using a nylon swab soaked in Amies buffer and incubated at 37°C for 24 h under shaking. TBs other than Amies were not tested since they contain carbon sources (Table S4), which would sustain bacterial growth during incubation at 37°C (Fig. S6), therefore invalidating a direct comparison with the resuscitation procedure. Except for *P. aeruginosa*, which was partially resuscitated, none of the ESKAPE bacteria regained cultivability in Amies buffer. Conversely, all Gram-negative ESKAPE species were almost completely resuscitated in RB (Fig. 2C; Fig. S7).

## Improved detection of the microbial load on dry surfaces through VBNC resuscitation

To assess the impact of the resuscitation step on bacterial quantification in a real-life environment, different surfaces within the Department of Science at Roma Tre University were sampled with three types of swab/buffer combinations, namely a nylon swab pre-moistened in RB, and the commercially available swabs LMS Swab Amies and ESC Swab MRD. Following surface sampling, an initial CFU count was performed ($T_0$). Subsequently, swabs were incubated for 24 h in RB at 37°C, or in Amies and MRD at 4°C ($T_1$) before CFU counting (Fig. 3A). CFU counts remained unchanged or even decreased

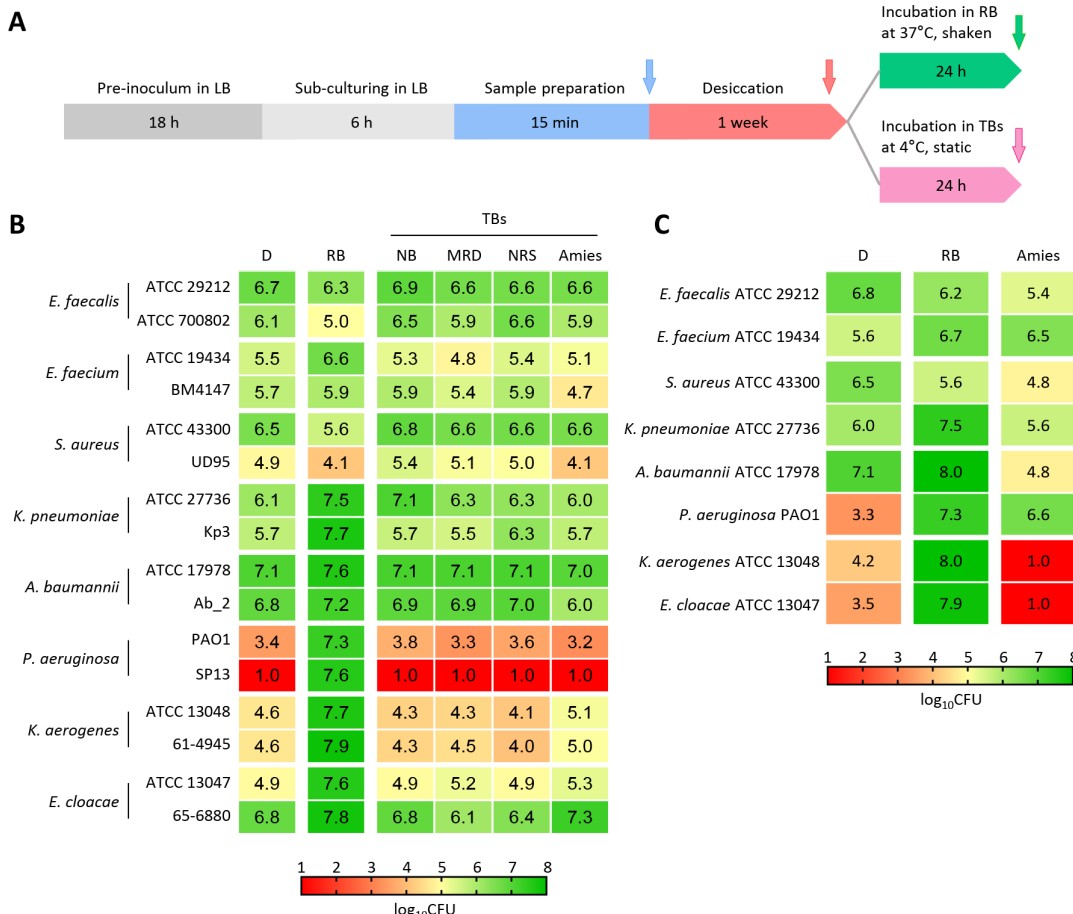

**FIG 2** Comparison between the standard procedure for environmental contamination control and the procedure based on the resuscitation process. (A) Timeline of experimental procedure. The experiment comprises four sequential steps: (i) bacterial cultivation (gray), (ii) sample preparation (blue), (iii) desiccation on glass slides (red), and (iv) resuscitation in RB at 37°C for 24 h (green) or incubation in commercial TBs at 4°C for 24 h under static conditions (pink). The arrows indicate the time points at which CFU counts were performed. (B) Cultivability of bacteria determined after 1-week desiccation (D lane) and subsequent 24-h incubation at 4°C for all the TBs, or after 24-h incubation at 37°C for RB, and expressed as $\log_{10}$ CFU. NB, neutralizing broth; MRD, maximum recovery diluent; NRS, neutralizing rinse solution. (C) Cultivability of ESKAPE bacteria after 1-week desiccation (D lanes) on glass slides and resuscitation in RB or Amies transport medium at 37°C for 24 h. Data in panels B and C are the means of three independent experiments. Mean values and error bars are shown in Fig. S4 and S6.

compared to the $T_0$ values for the commercial swabs, whereas incubation in RB markedly increased CFU counts, ranging from twofold up to $10^6$-fold relative to $T_0$ counts (Fig. 3B). These results suggest that conventional methods and commercially available swabs commonly used in surface sampling underestimate the actual bacterial load on abiotic surfaces.

## DISCUSSION

Prevention and control of HAI remain significant medical challenges. Pathogens responsible for HAI, particularly ESKAPE bacteria, can spread indirectly through contact with fomites (17, 18). More than 40% of HAI are due to cross-contamination by health-care personnel, who can transfer pathogens via contaminated cloths, gloves, and frequently touched surfaces such as medical devices, tables, and beds (19, 20). The ability of pathogens to persist on surfaces depends not only on surface properties, such as porosity, hydrophobicity, roughness, and intrinsic antimicrobial activities, but also on specific characteristics of the pathogen itself (21). Environmental persistence has been extensively documented in healthcare settings for some ESKAPE species, such as *K.*

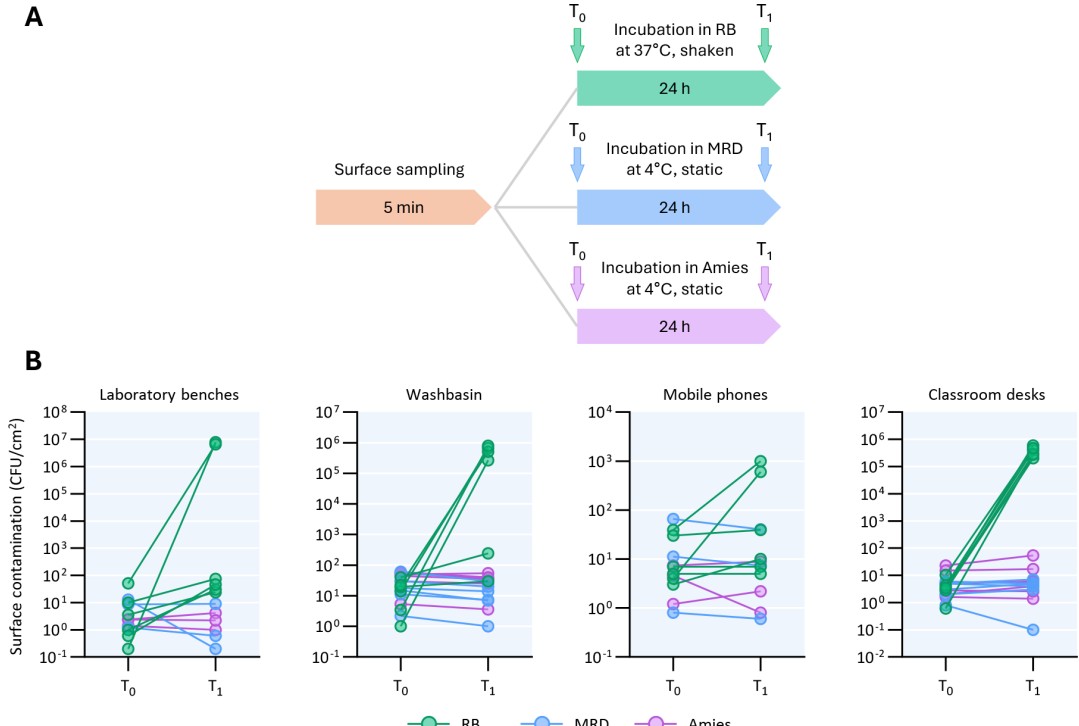

**FIG 3** Sampling of different surfaces for bacterial detection using commercial swabs and swabs soaked in the RB. (A) Timelines of surface sampling with nylon swab soaked with RB (green), MRD (blue), and Amies (purple). Laboratory benches, washbasins, mobile phones, and classroom desks were sampled using appropriate sampling masks and pre-moistened swabs. Swabs were incubated at room temperature for 15 min, vortexed for 1 min, and 100 µL were plated on TSA plates ($T_0$). Swabs immersed in RB were incubated at 37°C with shaking for 24 h, while those in Amies transport medium and MRD were stored at 4°C for 24 h before plating ($T_1$). (B) Surface contamination, expressed as the number of CFU per $cm^2$, determined at $T_0$ and $T_1$. For each surface type, six samples were collected and analyzed using RB, while three samples each were collected and analyzed using either Amies or MRD.

*pneumoniae, A. baumannii*, and *S. aureus*, as opposed to *P. aeruginosa,* which is believed to survive for a very short time in the dry nosocomial environment (8, 22). However, standard culture-dependent methods for the assessment of surface contamination in hospitals do not detect bacteria in the VBNC state, likely underestimating the actual microbial burden. Moreover, despite the clinical relevance of *K. aerogenes* and *Enterobacter cloacae* (23, 24), no studies have so far investigated the ability of these species to enter the VBNC state in response to environmental stresses. Here, we demonstrate that desiccation on different substrates induces the transition of both *K. aerogenes* and *E. cloacae* to a VBNC state, resulting in a marked loss of cultivability that can be reversed by the resuscitation process.

Since desiccation is known to cause severe membrane damage in bacteria due to lipid oxidation, membrane protein misfolding, loss of fluidity, and, ultimately, disruption of the lipid bilayer (11, 25), the LIVE/DEAD stain has in the past been proposed as a tool to identify VBNC cells (6). Given that SYTO 9 permeates both viable and dead cells, while PI only stains cells with compromised membranes, the LIVE/DEAD stain is predicted to differentiate between actively growing bacteria (SYTO 9-positive), VBNC cells (both SYTO 9- and PI-positive), and dead cells (only PI-positive). However, in line with recent findings (12, 13), here we demonstrate that even apparently dead cells exhibiting strong PI uptake, indicative of severe membrane damage, can be resuscitated. This observation suggests that membrane integrity alone cannot serve as a critical marker for VBNC detection, and that complete regain of membrane integrity by Gram-negative ESKAPE species is not mandatory for recovery of cultivability after resuscitation.

The emerging notion that a portion of the bacterial population exposed to air desiccation can enter the VBNC state and elude detection calls into question the

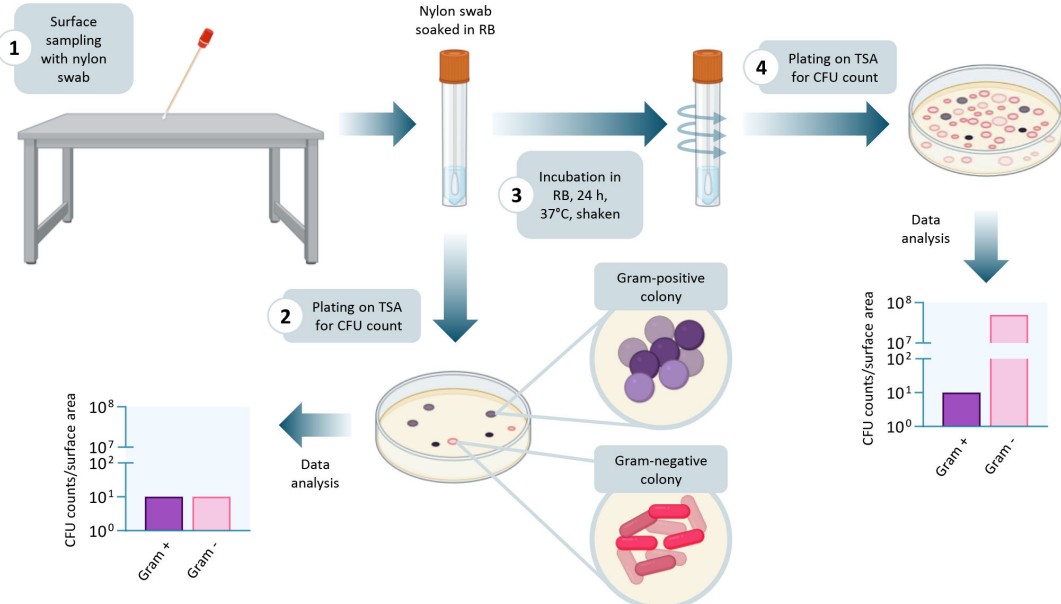

**FIG 4** Graphical representation of the surface sampling procedure with or without the resuscitation step. The direct plating of the transport medium in which the swab is immersed after surface sampling allows the culture-based detection of a minority of ESKAPE Gram-negative bacteria, i.e., those that do not enter the VBNC state. Incubation of the swab in RB for 24 h at 37°C under shaking enables ESKAPE Gram-negative bacteria in the VBNC state to regain cultivability, for a more reliable assessment of bacterial contamination of the surface. The figure was created with Biorender.

reliability of current environmental monitoring methods. In Gram-negative ESKAPE pathogens, entry into the VBNC state appears to be a widespread strategy to withstand the desiccation-induced stress. Our study shows that the substrate material can influence bacterial survival and highlights the remarkable ability of certain pathogens, particularly *P. aeruginosa*, to elude direct culture-based detection once they enter the VBNC state.

Remarkably, we demonstrate that a simple resuscitation step involving 24-h incubation at 37°C in RB is enough to entirely recover the colony-forming ability of otherwise uncultivable Gram-negative ESKAPE bacteria (Fig. 4). The limits of traditional methods have been confirmed by comparative tests of individual ESKAPE species exposed to defined dehydration conditions. The higher sensitivity of the resuscitation protocol over traditional methods was also corroborated by a pilot environmental sampling experiment.

However, some limitations of this work should be considered when interpreting our findings. Our study has focused on a restricted set of pathogenic species, representative of the ESKAPE group; therefore, it cannot be assumed that the resuscitation protocol will work with other clinically relevant species. In addition, although reference and clinical isolates of each ESKAPE species displayed similar behavior, the limited number of strains examined does not allow us to rule out potential intraspecific variability. Another critical point is that the procedure proved unsuccessful in resuscitating Gram-positive ESKAPE species. Although these species can enter a VBNC state in response to different stressors (Table S1), the resuscitation protocol did not increase their cultivability, suggesting either an irreversible loss of cultivability after desiccation or an inadequacy of the RB for these species. Furthermore, although the method proved effective for Gram-negative bacteria that entered a VBNC state after desiccation, it remains unclear whether the same procedure can resuscitate VBNC cells arising within biofilms or induced by other stressors, including antimicrobial compounds or disinfectants. Finally, the main limitation of this work is that our proposed protocol for environmental sampling has not yet been validated in healthcare settings, where factors such as the use of disinfectants, the presence of organic material from patients' biological fluids, and different temperature and humidity levels may further modulate bacterial responses.

Despite these limitations, our results clearly call into question the reliability of conventional methodologies for environmental surveillance (e.g., BS EN 17141:2020) and raise important issues about bacteria previously thought incapable of surviving desiccation on hospital surfaces. Microbiological methods for environmental surveillance should therefore be implemented to unveil the elusive bacterial species capable of thriving in the VBNC state. According to the BS EN 17141:2020 standard procedure suited for "Cleanrooms and associated controlled environments—Biocontamination control", environmental sampling can be performed using different devices, including contact plates, sponges, and swabs. Among these, the swab is the most frequently used device in healthcare settings (26, 27). Therefore, our protocol for surface sampling (Supplemental material and Fig. 4) can easily be implemented +in existing environmental monitoring procedures, requiring only an additional step, consisting of 24-h incubation at 37°C in RB.

## Conclusions

Based on our findings, we propose a simple method (protocol described on pages xviii–xix of the Supplemental material) for the recovery of bacteria on abiotic surfaces, which enables the detection of Gram-negative ESKAPE bacteria even in the VBNC state. The method is intended as an aid to infection control specialists to improve environmental surveillance standards.

## ACKNOWLEDGMENTS

D.V., Writing—original draft, Visualization, Validation, Supervision, Software, Resources, Project administration, Methodology, Investigation, Funding acquisition, Formal analysis, Data curation, Conceptualization | M.L., Writing—original draft, Visualization, Validation, Software, Resources, Methodology, Investigation, Data curation, Conceptualization | C.S., Methodology, Investigation, Data curation, Conceptualization | I.C., Methodology, Investigation, Data curation, Conceptualization | F.I., Writing—review and editing, Validation, Supervision, Resources, Methodology, Funding acquisition, Formal analysis | P.V., Writing—review and editing, Validation, Supervision, Resources, Methodology, Funding acquisition, Formal analysis.

## AUTHOR AFFILIATIONS

[1]Department of Science, Roma Tre University, Rome, Italy
[2]NBFC, National Biodiversity Future Center, Palermo, Italy
[3]Santa Lucia Foundation IRCCS, Rome, Italy

## AUTHOR ORCIDs

Daniela Visaggio 🔵 http://orcid.org/0000-0002-3192-8974
Massimiliano Lucidi 🔵 http://orcid.org/0000-0003-3238-9164
Francesco Imperi 🔵 http://orcid.org/0000-0001-5080-5665
Paolo Visca 🔵 http://orcid.org/0000-0002-6128-7039

## FUNDING

| Funder | Grant(s) | Author(s) |
| --- | --- | --- |
| Italian Ministry of Universities and Research (MUR) | Excellence Departments grant (art. 1, comma 314-337 Legge 232/2016) | Paolo Visca |
| Italian Ministry of Universities and Research (MUR) | PRIN 2022 grant (CUP F53D23000860001) | Daniela Visaggio |
| Italian Ministry of Universities and Research (MUR) | Rome Technopole, PNRR grant (M4-C2-Inv. 1.5 CUP F83B22000040006) | Paolo Visca |

## AUTHOR CONTRIBUTIONS

Daniela Visaggio, Conceptualization, Data curation, Formal analysis, Funding acquisition, Investigation, Methodology, Project administration, Resources, Software, Supervision, Validation, Visualization, Writing – original draft | Massimiliano Lucidi, Conceptualization, Data curation, Investigation, Methodology, Resources, Software, Validation, Visualization, Writing – original draft | Cinzia Spagnoli, Conceptualization, Data curation, Investigation, Methodology | Ilaria Ciccone, Conceptualization, Data curation, Investigation, Methodology | Francesco Imperi, Formal analysis, Funding acquisition, Methodology, Resources, Supervision, Validation, Writing – review and editing | Paolo Visca, Conceptualization, Formal analysis, Funding acquisition, Methodology, Resources, Supervision, Validation, Writing – review and editing

## ETHICS APPROVAL

All experiments were conducted in accordance with institutional and national biosafety regulations. The clinical isolates were either previously published, as indicated in the references listed in Table S2, or obtained anonymously from the biobank of a regional hospital.

## ADDITIONAL FILES

The following material is available online.

### Supplemental Material

**Supplemental material (Spectrum03357-25-s0001.pdf).** Tables S1 to S4, Fig. S1 to S7, and supplemental methods.

### Open Peer Review

**PEER REVIEW HISTORY (review-history.pdf).** An accounting of the reviewer comments and feedback.

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
