## [Reviewer comments · Microbiology Spectrum]

Microbiology Spectrum

ESKAPE Gram-negative bacteria escape culture-based detection upon desiccation on abiotic surfaces

Daniela Visaggio, Massimiliano Lucidi, Cinzia Spagnoli, Ilaria Ciccone, Francesco Imperi, and Paolo Visca

Corresponding Author(s): Paolo Visca, Universita degli Studi Roma Tre Dipartimento di Scienze

Review Timeline:

Submission Date:	October 20, 2025
Editorial Decision:	November 17, 2025
Revision Received:	December 3, 2025
Accepted:	January 5, 2026

Editor: Bobby Warren

Reviewer(s): Disclosure of reviewer identity is with reference to reviewer comments included in decision letter(s). The following individuals involved in review of your submission have agreed to reveal their identity: Hajime Kanamori (Reviewer #1)

Transaction Report:

DOI: <https://doi.org/10.1128/spectrum.03357-25>

Re: Spectrum03357-25 (ESKAPE Gram-negative bacteria escape culture-based detection upon desiccation on abiotic surfaces)

Dear Prof. Paolo Visca:

Thank you for the privilege of reviewing your work. Below you will find my comments, instructions from the Spectrum editorial office, and the reviewer comments.

Revision Guidelines

Sincerely,
Bobby Warren
Editor
Microbiology Spectrum

Reviewer #1 (Comments for the Author):

Thank you for providing me with the opportunity to review this manuscript: ESKAPE Gram-negative bacteria escape culture-based detection upon desiccation on abiotic surfaces. The authors investigated the survival and detection of ESKAPE pathogens on abiotic surfaces under desiccation stress. The study demonstrated that Gram-negative ESKAPE bacteria can enter VBNC state upon desiccation and resuscitate in a carbon-

free buffer which step increased detection sensitivity in environmental samples. This study also revealed that conventional culture-based environmental monitoring underestimates surface contamination. This manuscript is well-written, but some points need to be improved.

1. The VBNC state under desiccation is convincingly shown, but applying the data to real hospital environments (e.g., variable humidity, organic load, cleaning agents, biofilms) remains unclear. Please discuss this further.
2. The resuscitation buffer has an important role in this study. Please detail its composition and rationale (carbon-free, isotonic, not growth-permissive) to facilitate adoption by other laboratories and healthcare settings. The resuscitation step increases sensitivity but may also overestimate detection counts due to growth of contaminants. Would the authors clarify this?
3. Please include a short statement on biosafety and ethical considerations for clinical strains and environmental sampling.

Reviewer #2 (Comments for the Author):

Major Concerns:

• **Methods Organization:**

The Materials and Methods section is detailed and informative, but could be improved by clearer subheadings (e.g., "Bacterial Strains," "Desiccation Protocol," "Resuscitation Conditions," "Environmental Sampling," etc.) for readability.

• **Experimental Controls:**

The study could better explain the rationale behind using specific materials. Were these chosen for prevalence in clinical settings or for other reasons? Clarifying this would strengthen the experimental design justification.

• **Statistical Analysis:**

While data are presented with mean values and figures, there's no mention of statistical tests to determine significance between conditions (e.g., pre- and post-resuscitation, surface types). Including the statistical methods used (or adding them if not done) is essential.

• **Environmental Sampling Interpretation:**

In the Results, the environmental sampling experiment demonstrates improved detection using RB, but the Discussion could elaborate on the potential clinical implications-how would this revised approach be incorporated into existing hospital surveillance systems?

• **Discussion - Broader Context and Limitations:**

The Discussion does a good job summarizing findings but could be strengthened by:

- Adding a limitations subsection
- Expanding on how these findings translate into practical infection control procedures or how they might influence updates to standards.

• **Figures and Supplementary Data:**

The text references multiple supplementary tables and figures. Consider integrating key data (e.g., CFU differences before and after resuscitation) into the main figures for easier interpretation, reserving less critical data for the supplement.

Minor Concerns:

• **Abstract:**

The phrase "can be resuscitated in a carbon-free buffer" could be clarified to emphasize that the bacteria regained cultivability but not necessarily full physiological activity.

Also, "resulted in much higher recovery" could be rephrased to "yielded a significantly greater recovery" for scientific precision.

• **Introduction:**

The transition from the general concept of VBNC states to the focus on ESKAPE pathogens could be smoother by briefly noting their importance in nosocomial infections before listing the species.

• **LIVE/DEAD Staining Description:**

Line 135 mentions a "custom macro script" for ImageJ-this should be briefly summarized or cited for reproducibility.

• **Typographical and Formatting Issues:**

A few minor typographical errors (e.g., "detecs" instead of "detects" in line 83) should be corrected. Ensure consistency in terms like "resuscitation buffer (RB)" and "transport buffer (TB)."

• **Conclusion Section:**

The Discussion's final paragraph effectively summarizes the practical implications. However, it could be explicitly labeled as a "Conclusion" to improve readability and align with journal formatting expectations.

Summary

The authors present a study evaluating the persistence and detection of ESKAPE Gram-negative pathogens on abiotic surfaces after desiccation. They demonstrate that these bacteria can enter a viable but non-culturable (VBNC) state, allowing them to evade standard culture-based detection methods. By introducing a resuscitation step using a carbon-free buffer (RB), they show significantly improved recovery rates for Gram-negative ESKAPE bacteria. Their findings highlight a critical limitation in current environmental monitoring protocols and propose an enhanced method for identifying VBNC pathogens on hospital surfaces.

Major Concerns

- **Methods Organization:**
The Materials and Methods section is detailed and informative, but could be improved by clearer subheadings (e.g., “Bacterial Strains,” “Desiccation Protocol,” “Resuscitation Conditions,” “Environmental Sampling,” etc.) for readability.
- **Experimental Controls:**
The study could better explain the rationale behind using specific materials. Were these chosen for prevalence in clinical settings or for other reasons? Clarifying this would strengthen the experimental design justification.
- **Statistical Analysis:**
While data are presented with mean values and figures, there’s no mention of statistical tests to determine significance between conditions (e.g., pre- and post-resuscitation, surface types). Including the statistical methods used (or adding them if not done) is essential.
- **Environmental Sampling Interpretation:**
In the Results, the environmental sampling experiment demonstrates improved detection using RB, but the Discussion could elaborate on the potential clinical implications—how would this revised approach be incorporated into existing hospital surveillance systems?
- **Discussion – Broader Context and Limitations:**
The Discussion does a good job summarizing findings but could be strengthened by:
 - Adding a limitations subsection
 - Expanding on how these findings translate into practical infection control procedures or how they might influence updates to standards.
- **Figures and Supplementary Data:**
The text references multiple supplementary tables and figures. Consider integrating key data (e.g., CFU differences before and after resuscitation) into the main figures for easier interpretation, reserving less critical data for the supplement.

Minor Concerns

- **Abstract:**
The phrase “can be resuscitated in a carbon-free buffer” could be clarified to emphasize that the bacteria regained *cultivability* but not necessarily full physiological activity.
Also, “resulted in much higher recovery” could be rephrased to “yielded a significantly greater recovery” for scientific precision.
- **Introduction:**
The transition from the general concept of VBNC states to the focus on ESKAPE pathogens could

be smoother by briefly noting their importance in nosocomial infections *before* listing the species.

- LIVE/DEAD Staining Description:
Line 135 mentions a “custom macro script” for ImageJ—this should be briefly summarized or cited for reproducibility.
- Typographical and Formatting Issues:
A few minor typographical errors (e.g., “detecs” instead of “detects” in line 83) should be corrected. Ensure consistency in terms like “resuscitation buffer (RB)” and “transport buffer (TB).”
- Conclusion Section:
The Discussion’s final paragraph effectively summarizes the practical implications. However, it could be explicitly labeled as a “Conclusion” to improve readability and align with journal formatting expectations.

Reviewer #1 (Comments for the Author):

Thank you for providing me with the opportunity to review this manuscript: ESKAPE Gram-negative bacteria escape culture-based detection upon desiccation on abiotic surfaces.

The authors investigated the survival and detection of ESKAPE pathogens on abiotic surfaces under desiccation stress. The study demonstrated that Gram-negative ESKAPE bacteria can enter VBNC state upon desiccation and resuscitate in a carbon-free buffer which step increased detection sensitivity in environmental samples. This study also revealed that conventional culture-based environmental monitoring underestimates surface contamination. This manuscript is well-written, but some points need to be improved.

1. The VBNC state under desiccation is convincingly shown, but applying the data to real hospital environments (e.g., variable humidity, organic load, cleaning agents, biofilms) remains unclear. Please discuss this further.

R1-1: We appreciate the Reviewer's insightful comment regarding the applicability of our findings to real hospital environments. Although the experiments were performed in a real-life university environment rather than in hospital wards, our preliminary findings should be considered a proof of concept. Indeed, some factors such as variable humidity, organic loads, and disinfectant use can differ not only between university and hospital settings but also among hospital settings. Although additional research under hospital-specific conditions is needed (and it is planned among our future activities), our results suggest that resuscitation could play a key role in shaping microbial persistence and detection in clinical settings.

This relevant concept is explicitly addressed in the Discussion section (lines 309–312).

2. The resuscitation buffer has an important role in this study. Please detail its composition and rationale (carbon-free, isotonic, not growth-permissive) to facilitate adoption by other laboratories and healthcare settings. The resuscitation step increases sensitivity but may also overestimate detection counts due to the growth of contaminants. Would the authors clarify this?

R1-2: In the original version of the manuscript, the composition of the buffer was provided in Table S4. According to the Reviewer's suggestion, we have now also reported the buffer composition in the Materials and Methods section of the revised manuscript (lines 99–100). In the Results section we have further clarified that the buffer does not support bacterial growth due to the absence of any carbon source (lines 195–196). This means that the resuscitation buffer allows VBNC cells to recover their culturability when plated on nutrient agar media but cannot promote bacterial proliferation by itself, as experimentally confirmed by the growth assays shown in Figure S2. This excludes that the results could be affected by bacterial growth in the resuscitation buffer. Please also note that the different surfaces tested in the in vitro experiments (Figures 1 and 2) were sterilized prior to use, and that these experiments were conducted under aseptic conditions, thus ruling out the presence of contaminants.

3. Please include a short statement on biosafety and ethical considerations for clinical strains and environmental sampling.

R1-3: Following the Reviewer's suggestion, we have added a short statement on biosafety and ethical considerations in the Materials and Methods section (lines 175–178).

Reviewer #2 (Comments for the Author):

Major Concerns:

We express our gratitude to the Reviewer for her/his careful revision of our manuscript.

• Methods Organization:

The Materials and Methods section is detailed and informative, but could be improved by clearer subheadings (e.g., "Bacterial Strains," "Desiccation Protocol," "Resuscitation Conditions," "Environmental Sampling," etc.) for readability.

R2-1: In accordance with the Reviewer's suggestion, the Materials and Methods section has been revised to include subheadings (line 84 and line 97).

• Experimental Controls:

The study could better explain the rationale behind using specific materials. Were these chosen for prevalence in clinical settings or for other reasons? Clarifying this would strengthen the experimental design justification.

R2-2: The materials used in this study were selected to reflect the variety of surfaces and devices commonly found in hospital environments. Table S3 of the original (and revised) manuscript provides some examples of hospital devices for each material used in this study. For further clarity, a sentence has now been added to the Materials and Methods section to explain the rationale behind the selection of these materials (lines 92-93).

• Statistical Analysis:

While data are presented with mean values and figures, there's no mention of statistical tests to determine significance between conditions (e.g., pre- and post-resuscitation, surface types).

Including the statistical methods used (or adding them if not done) is essential.

R2-3: We thank the Reviewer for the valuable comment. Following the Reviewer's suggestion, we performed statistical analyses as indicated in the Materials and Methods section (lines 170-173) and in the figure legends of Figures S1, S4 (S5 in the revised manuscript), and S6 (S7 in the revised manuscript). In addition, we have included a new supplementary figure (Figure S2) that highlights the statistically significant differences in cultivability after the assay across the different surfaces.

• Environmental Sampling Interpretation:

In the Results, the environmental sampling experiment demonstrates improved detection using RB, but the Discussion could elaborate on the potential clinical implications-how would this revised approach be incorporated into existing hospital surveillance systems?

R2-4: We thank the Reviewer for this insightful comment. We believe that improved detection of VBNC bacteria through our resuscitation protocol we propose has clear potential implications for hospital infection control. Detecting Gram-negative hospital pathogens even in the VBNC state could provide a more accurate assessment of environmental contamination and help guide targeted cleaning and disinfection strategies, ultimately supporting infection prevention efforts. Surface sampling using swabs is currently the most commonly used and accurate sampling

procedure (Rawlinson et al., 2019; Chen et al., 2024), and therefore, our approach could be integrated into existing hospital environmental surveillance programs with minimal modifications, requiring only a 24-hour incubation step at 37°C. We have now improved the Discussion to highlight these points (lines 317–323).

• Discussion - Broader Context and Limitations:

The Discussion does a good job summarizing findings but could be strengthened by:

- Adding a limitations subsection

R2-5: We thank the Reviewer for this helpful suggestion. We have now expanded the Discussion to discuss the main limitations of our work (lines 297–312).

- Expanding on how these findings translate into practical infection control procedures or how they might influence updates to standards.

R2-6: Following the Reviewer's suggestion, we have expanded the Discussion to highlight how the VBNC detection method could improve infection control practices (lines 325-328).

• Figures and Supplementary Data:

The text references multiple supplementary tables and figures. Consider integrating key data e.g., CFU differences before and after resuscitation) into the main figures for easier interpretation, reserving less critical data for the supplement.

R2-7: The supplemental material shows (and provides statistical analysis) of CFU counts before desiccation, after desiccation, and after resuscitation in three figures (Figures S1, S2, and S3), comprising a total of 43 panels. Authors made considerable effort to condense this large amount of information into two main figures with 12 panels (9 panels in Figure 1 and 3 panels in Figure 2). To facilitate reading, Figures 1 and 2 employ an intuitive heat map format. The color coding provides an immediate visual overview of CFU dynamics following desiccation and resuscitation, while the Log_{10} CFU values are included in the figures.

Minor Concerns:

• Abstract:

The phrase "can be resuscitated in a carbon-free buffer" could be clarified to emphasize that the bacteria regained cultivability but not necessarily full physiological activity.

R2-8: The concept of resuscitation, understood as the recovery of cultivability, has been expressed on lines 23–25, and we believe that repeating it is therefore unnecessary.

Also, "resulted in much higher recovery" could be rephrased to "yielded a significantly greater recovery" for scientific precision.

R2-9: The text has been revised according to the Reviewer's suggestions (lines 33-34).

• Introduction:

The transition from the general concept of VBNC states to the focus on ESKAPE pathogens could be smoother by briefly noting their importance in nosocomial infections before listing the species.

R2-10: We thank the Reviewer for this suggestion; the importance of ESKAPE pathogens in healthcare-associated infections is described immediately after the name of the species (lines 76-78).

• LIVE/DEAD Staining Description:

Line 135 mentions a "custom macro script" for ImageJ-this should be briefly summarized or cited for reproducibility.

R2-11: The custom macro script for CLSM image processing is reported in the Supplementary Materials.

• Typographical and Formatting Issues:

A few minor typographical errors (e.g., "detecs" instead of "detects" in line 83) should be corrected. Ensure consistency in terms like "resuscitation buffer (RB)" and "transport buffer (TB)."

R2-12: The typographical error indicated by the Reviewer has been corrected (e.g., 'detecs' changed to 'detects' on line 82). Following the Reviewer's suggestion, we have now written 'resuscitation buffer (RB)' and 'transport buffer (TB)' in full the first time they appear in the text. In the figure legends, both the full term and the abbreviation are provided to make the figures easily readable.

• Conclusion Section:

The Discussion's final paragraph effectively summarizes the practical implications. However, it could be explicitly labeled as a "Conclusion" to improve readability and align with journal formatting expectations.

R2-13: As suggested by the Reviewer, the final paragraph of the Discussion has been explicitly labeled as 'Conclusion' (lines 324-328) to improve readability and align with journal formatting expectations.

Re: Spectrum03357-25R1 (ESKAPE Gram-negative bacteria escape culture-based detection upon desiccation on abiotic surfaces)

Dear Prof. Paolo Visca:

Your manuscript has been accepted, and I am forwarding it to the ASM production staff for publication. Your paper will first be checked to make sure all elements meet the technical requirements. ASM staff will contact you if anything needs to be revised before copyediting and production can begin. Otherwise, you will be notified when your proofs are ready to be viewed.

Sincerely,
Bobby Warren
Editor
Microbiology Spectrum